# Prediction of Neonatal Respiratory Morbidity Assessed by Quantitative Ultrasound Lung Texture Analysis in Twin Pregnancies

**DOI:** 10.3390/jcm11164895

**Published:** 2022-08-20

**Authors:** Ana L. Moreno-Espinosa, Ameth Hawkins-Villarreal, David Coronado-Gutierrez, Xavier P. Burgos-Artizzu, Raigam J. Martínez-Portilla, Tatiana Peña-Ramirez, Dahiana M. Gallo, Stefan R. Hansson, Eduard Gratacòs, Montse Palacio

**Affiliations:** 1BCNatal-Fetal Medicine Research Center, Hospital Clínic and Hospital Sant Joan de Déu, Universitat de Barcelona, 08028 Barcelona, Spain; 2Department of Obstetrics and Gynecology, Hospital Santo Tomás, Universidad de Panamá, Panama City 07096, Panama; 3Iberoamerican Research Network in Obstetrics, Gynecology and Translational Medicine, Mexico City 06720, Mexico; 4Transmural Biotech SL, 08021 Barcelona, Spain; 5Clinical Research Branch, National Institute of Perinatology, Mexico City 11000, Mexico; 6School of Medicine, Universidad del Valle, Cali 760032, Colombia; 7Department of Obstetrics and Gynecology, Hospital Universitario del Valle Evaristo García E.S.E., Cali 760043, Colombia; 8Department of Obstetrics and Gynecology, Institute of Clinical Sciences Lund, Lund University, 221 00 Lund, Sweden; 9Skåne University Hospital, 214 28 Malmö, Sweden; 10Centre for Biomedical Research on Rare Diseases (CIBERER), 28029 Madrid, Spain; 11Institut d’Investigacions Biomèdiques August Pi i Sunyer, 08036 Barcelona, Spain

**Keywords:** fetal lung maturity, quantitative lung texture analysis, twin pregnancies, neonatal respiratory morbidity

## Abstract

The objective of this study was to evaluate the performance of quantitative ultrasound of fetal lung texture analysis in predicting neonatal respiratory morbidity (NRM) in twin pregnancies. This was an ambispective study involving consecutive cases. Eligible cases included twin pregnancies between 27.0 and 38.6 weeks of gestation, for which an ultrasound image of the fetal thorax was obtained within 48 h of delivery. Images were analyzed using quantusFLM^®^ version 3.0. The primary outcome of this study was neonatal respiratory morbidity, defined as the occurrence of either transient tachypnea of the newborn or respiratory distress syndrome. The performance of quantusFLM^®^ in predicting NRM was analyzed by matching quantitative ultrasound analysis and clinical outcomes. This study included 166 images. Neonatal respiratory morbidity occurred in 12.7% of cases, and it was predicted by quantusFLM^®^ analysis with an overall sensitivity of 42.9%, specificity of 95.9%, positive predictive value of 60%, and negative predictive value of 92.1%. The accuracy was 89.2%, with a positive likelihood ratio of 10.4, and a negative likelihood ratio of 0.6. The results of this study demonstrate the good prediction capability of NRM in twin pregnancies using a non-invasive lung texture analysis software. The test showed an overall good performance with high specificity, negative predictive value, and accuracy.

## 1. Introduction

The reported rates of neonatal respiratory morbidity (NRM) in twins are variable. Overall rates of 13.5% [1] to 19% [2] have been addressed in different studies. In addition, rates of 5.34% have been described for late preterm twins [3]. Gender [4], birth order [5], chorionicity [3], and birthweight discordance [6,7] are some of the factors which may affect the risk of NMR. In twin pregnancies, there is usually disparity between twins, so the risk of respiratory morbidity may be different for each infant in each twin pair [8].

Non-invasive prediction of fetal lung maturity by ultrasound has been under research for many years now. The method used by quantusFLM^®^ is based on a combination of machine learning and texture extraction, which have shown a strong correlation with gestational age [9], and a test to predict fetal lung maturation (FLM), previously widely performed on amniotic fluid [10]. The prediction of NRM was also addressed in a single-center [11] and a multicenter study [12], in which pregnancies with different co-morbidities were recruited, including multiple pregnancies.

The prevalence of twins born at <37 weeks is almost invariably reported to be around 50%, with 32% of births prior to 35 weeks, between 15 and 20% at <34 weeks, and 9 to 11.4% at <32 weeks [13,14,15,16,17].

Publications on NRM and its prediction in twin pregnancies are scarce and present mixed outcomes, with twin pregnancies frequently being an exclusion criterion to evaluate results in diagnostic tests. Some studies have evaluated fetal lung maturity in twin pregnancies [18,19,20]; however, none of these studies have focused on predicting NRM or reporting the performance of the tests used. We hypothesized that the performance of quantusFLM^®^ in predicting NRM in neonates from twin pregnancies would be comparable to that previously reported in the general population.

The objective of this study was to evaluate the performance of quantitative ultrasound fetal lung texture analysis in predicting NRM in twin pregnancies.

## 2. Materials and Methods

### 2.1. Patient Recruiting

This was an ambispective study involving twin pregnancies. Prospective cases were recruited at the Hospital Clinic, Barcelona, Spain, in collaboration with Hospital Universitario del Valle, “Evaristo García” E.S.E., Cali, Colombia. Patients were recruited from January 2018 to February 2020. Retrospective cases were identified from a database designed for a multicenter study [12] (recruited from June 2011 to December 2014), and the information was added to the present study for the analysis. Eligible cases included consecutive cases of twin pregnancies between 25.0 and 38.6 weeks of gestation, for which an ultrasound image of the fetal thorax was obtained within 48 h of delivery. Every ultrasound image of the fetal thorax included in the study for analysis corresponded to a fetus from a twin pregnancy. When the image of one twin could not be obtained or was discarded after quality control, the image of the remaining twin was included. Fetal position was the main factor that did not allow for obtainment of the image, regardless of the presentation or whether it was the first or second fetus. Images were discarded after image quality control when the following were present: insufficient magnification, blurred images, calipers within the area of analysis, and acoustic shadows. Cases of twin pregnancies in which fetal death occurred spontaneously or secondary to procedures such as placental laser due to twin-to-twin transfusion syndrome, cord occlusion due to fetal malformation, or selective intrauterine growth restriction were also included.

Cases were excluded in two scenarios: (i) if steroids for fetal lung maturity had been used between the image acquisition and delivery, and (ii) fetuses with known congenital malformations or chromosomal abnormalities. Additionally, in the postnatal period, we excluded: neonates with sepsis, umbilical artery pH < 7.00, symptomatic anemia, postnatal diagnosis of chromosomal abnormalities, and meconium aspiration syndrome, which could explain respiratory difficulties due to reasons other than lung immaturity.

### 2.2. Image Processing

Ultrasound images were obtained following an acquisition protocol as detailed previously [12]. Images fulfilling the quality criteria were uploaded via the Internet by the engineers at the coordinating center through restricted access to a commercial software website (www.quantusflm.com (accessed on 26 February 2021); Transmural Biotech, Barcelona, Spain) and analyzed using the new quantusFLM^®^ version 3.0 [21]. This software automatically delineates a region of interest (ROI) in the fetal lung and calculates an NRM risk score, defined as the occurrence of either respiratory distress syndrome (RDS) or transient tachypnea of the newborn (TTN), as a continuous variable. To evaluate the risk of NRM, continuous output NRM risk scores were binarized using the optimal cut-off point threshold, computed as that which maximizes accuracy in the test images, thereby obtaining a categorical result (i.e., high or low risk). The optimal cut-off threshold was computed as that which maximizes the F1-Score using the entire dataset. The F1-Score is an accuracy metric which balances sensitivity and positive predictive values (PPV) to better judge the real usefulness of the prediction. It measures the harmonic average between sensitivity and PPV and is defined as (2 × True Positives)/(2 × True positives + False Positives + False Negatives).

### 2.3. Reference Standard

The primary outcome of the study was the development of NRM defined as the occurrence of either TTN or RDS. Perinatal and neonatal characteristics and outcomes were recorded from clinical charts in databases designed for the study. RDS was defined based on the typical chest radiography findings and admission to the neonatal intensive care unit for respiratory support, or the need for supplemental oxygen, together with clinical criteria, including grunting, nasal flaring, tachypnea, and chest wall retraction. Transient tachypnea of the newborn was diagnosed based on a chest X-ray showing hyperaeration of the lungs and prominent pulmonary vascular patterns, together with the clinical criteria of early and short-lived respiratory distress (isolated tachypnea, rare grunting, minimal retraction). 

### 2.4. Ethical Approval

All patients included in the study provided written informed consent for the use of ultrasound images and perinatal data. The study was approved by the Institutional Review Board of the Hospital Clinic of Barcelona (HCB/2017/0642), and Hospital Universitario del Valle, “Evaristo García” E.S.E. (008-2019).

### 2.5. Statistical Analysis

Quantitative variables were assessed using the Shapiro–Wilk test for normality, and normally distributed variables were expressed as the mean and standard deviation (SD). Non-normally distributed quantitative variables were expressed as the median and interquartile range (IQR: p25–75). Qualitative variables were reported as frequencies and percentages. The performance of quantusFLM^®^ in predicting NRM was analyzed by cross-tabulation of the results of the test against those of the reference standard, in this case the neonatal diagnosis of TTN or RDS. Contingency tables were used to estimate accuracy measurements. Fagan nomograms were constructed to show the pre-test probabilities of NRM in twins at different gestational ages, positive and negative likelihood ratios, and post-test probabilities. The data were analyzed using STATA, v.15.0 (College Station, TX, USA).

## 3. Results

### 3.1. Patient and Sample Characteristics

**Prospective data**: A total of 102 images were acquired, 10 (9.8%) of which were discarded after image quality control and 13 (12.7%) of which were excluded due to the use of antenatal steroids between the image acquisition and delivery. The remaining 79 images were included in the study. **Retrospective data**: A total of 87 images already chosen in the selection process of a previous study were added for analysis. This study included 166 images from 166 fetuses which were stratified into three groups: from 25.0 to 33.6 weeks [35/166 (21.1%)], from 34.0 to 36.6 weeks [48/166 (28.9%)], and from 37.0 to 38.6 weeks [83/166 (50.0%)]. The flowchart of the eligible cases according to the Standards for Reporting of Diagnostic Accuracy Studies (STARD) guidelines is shown in Figure 1. The excluded cases with the quantusFLM^®^ results and perinatal outcomes are shown in Appendix A.

The baseline and clinical characteristics of the women included in the study are described in Table 1. This study included 1 woman at <28.0 weeks; 20 women at 28.0 to <34.0 weeks; 32 women at 34.0 to <37.0 weeks; and 52 women at ≥37.0 weeks of gestation. Perinatal and neonatal characteristics are shown in Table 2. The overall prevalence of NRM was 12.7% (21/166), of which 61.9% (13/21) was diagnosed as TTN and 38.1% (8/21) as RDS. The prevalence of NRM was 42.9% (15/35) in the group from 25.0 to 33.6 weeks, and 12.5% (6/48) from 34.0 to 36.6 weeks, while in the group > 37.0 weeks, we found no cases of NRM. The newborns diagnosed with TTN received continuous positive airway pressure (CPAP) [92.3% (12/13)], oxygen ≥ 40% [23.1% (3/13)], or non-invasive ventilation/bilevel positive airway pressure (NIV/BIPAP) [15.4% (2/13)] up to 48 h. While newborns diagnosed with RDS were treated with invasive ventilation [62.5% (5/8)] or a combination of CPAP, NIV/BIPAP, and oxygen ≥ 40% [37.5% (3/8)], almost all received at least one dose of surfactant [87.5% (7/8)] and all were born below 34.0 weeks.

### 3.2. Performance of the Test

The occurrence of NRM was predicted by the quantusFLM^®^ analysis in the overall population with a sensitivity of 42.9% (9/21), specificity of 95.9% (139/145), positive predictive value (PPV) of 60% (9/15), and negative predictive value (NPV) of 92.1% (139/151). The accuracy was 89.2% (148/166), the positive likelihood ratio (LR+) was 10.4, and the negative likelihood ratio (LR-) was 0.6. Table 3 shows the performance of the tests by groups of gestational age. A summary of the overall performance of quantusFLM^®^ described in the general population in previous studies and in twin pregnancies is shown in Appendix A.

Figure 2 depicts the Fagan nomogram analysis to evaluate the clinical utility of the prediction of neonatal respiratory morbidity by quantusFLM^®^ in twin pregnancies. Appendix A shows the pre-test and post-test risk and probabilities of neonatal respiratory morbidity in twins.

## 4. Discussion

In this study, we explored, for the first time, the performance of a non-invasive lung texture analysis in predicting NRM in twin pregnancies. Considering different ranges of gestational age, we found better performance below 34.0 weeks with a specificity of 97.6%, an NPV of 73.4%, and an LR+ of 22.3, allowing for the identification of fetuses at high risk of NRM, with an accuracy of 78.4%. A high-risk result is accurate in predicting the presence of NRM because the LR for this result generates large changes in the pre-test probabilities of NRM. In the group between 34.0 and 36.6 weeks, we found a specificity of 85.7%, an NPV of 87.8%, and an LR+ of 1.2, albeit with a low sensitivity of 16.7%. A high-risk result generates fewer changes in the pre-test probabilities, and a low-risk result is less accurate in predicting the absence of NRM, as shown in the Fagan plots. Above 37 weeks, there was no case of RDS/TTN, precluding the calculation of all the parameters. All images were classified as having a low risk of NRM. Therefore, the ability of the test to correctly identify a fetus without NRM with a negative result is effective at any gestational age.

Compared to the results previously reported for the test in the general population, the twin results showed a similar performance in the overall specificity, with 94.7% and 95.9%, respectively. The negative predictive value (NPV) and accuracy changed from 95.4% to 92.1% and 91.5% to 89.2%, respectively. A more pronounced decrease was noted in the sensitivity from 71.0% to 42.9%, the PPV from 67.9 to 60%, and the F1-Score from 69.4 to 50.0%.

These findings are in line with those reported by Tsuda et al. [23], who evaluated a model combining gestational age and lamellar body count (LBC) to predict NRM in twins. They reported a sensitivity of 69.0% and a specificity of 88.0% for the best cut-off value, which is in line with our results in the gestational age group <32 weeks, and obtained a specificity of 97.0% for predicting RDS/TTN, although in the gestational age group >37 weeks, the diagnostic accuracy decreased compared to the preterm period. Gestational age is the strongest factor associated with FLM and should therefore be considered in the interpretation of a false positive and/or a false negative result.

The overall prevalence of NRM in our population was 12.7%, which is in line with that reported by Tsuda et al., using the amniotic lamellar body count (LBC) as a predictive tool in different series of twin pregnancies [1,23]. Other studies have reported prevalence of up to 19% including TTN and RDS [2].

The management of twin pregnancies remains challenging, even more so when the delivery of preterm fetuses is indicated due to medical conditions. As 32% of twins have been reported to be born prior to 35 weeks, in this scenario, more than 30% of twins would receive antenatal steroids, considering that the most widespread use of antenatal steroids is up to 34.0/34.6 weeks. However, harmful effects should be considered. Many studies have shown that the administration of corticosteroids in twin pregnancies does not improve neonatal morbidity and mortality [24], but rather can cause higher rates of hypoglycemia [25] or reduced fetal biometry [26]. It should also be noted that nearly 75% of women with twins deliver outside the optimal window for either the initial or rescue corticosteroid courses [27]. 

Given the controversial data that do not clearly show the same benefits of corticosteroids in twin pregnancies compared to singletons, and the increasing data showing that there may even be harmful effects [28,29], the prediction of NRM may play a role in the decision-making process. Its usefulness could also be tested in clinical protocols when corticosteroids have already been administered and an attempt is made to avoid repeated doses. Additionally, the technique can be used in any center in the world, and reliable results can be obtained if good-quality images are sent via the web for analysis.

The main strength of our study is that the prediction of NRM of fetuses from twin pregnancies was evaluated with a non-invasive, machine learning-based technology. This technology has proven to be robust in the general population and has the advantages of being accessible and easy to use. In the present study, only 10 images were discarded after image quality control. The method tested herein is an indirect approach to predict NRM, and it is largely related to gestational age, but not to other factors influencing lung maturity status. Our study had a limited number of cases with NRM in some gestational ages compared to late gestational ages, in which NRM is a rarer event. However, the Fagan plots showed that, although there were fewer changes compared to the pre-test probability, the method can provide useful information if needed, and the NPV supports the strategy in ruling out NRM. Additionally, the algorithms have not been designed for each specific gestational age, precluding assessment of the performance of the software in each gestational age. However, the gestational age range is a widely used measure to drive clinical decisions in the field of maternal–fetal medicine.

In summary, the results of this study show that NRM in twins can be predicted by a non-invasive lung texture analysis with an overall good specificity, NPV, and accuracy. QuantusFLM^®^ may be useful in planning indicated delivery of twin pregnancies because of medical conditions and may help to avoid repeated doses of corticosteroids when the fetuses have already been exposed and the risk of preterm delivery is still present. Therefore, in adequate facilities, this technology can be incorporated into protocols according to gestational age and may be helpful in the decision-making process when delivery is planned.

## Figures and Tables

**Figure 1 jcm-11-04895-f001:**
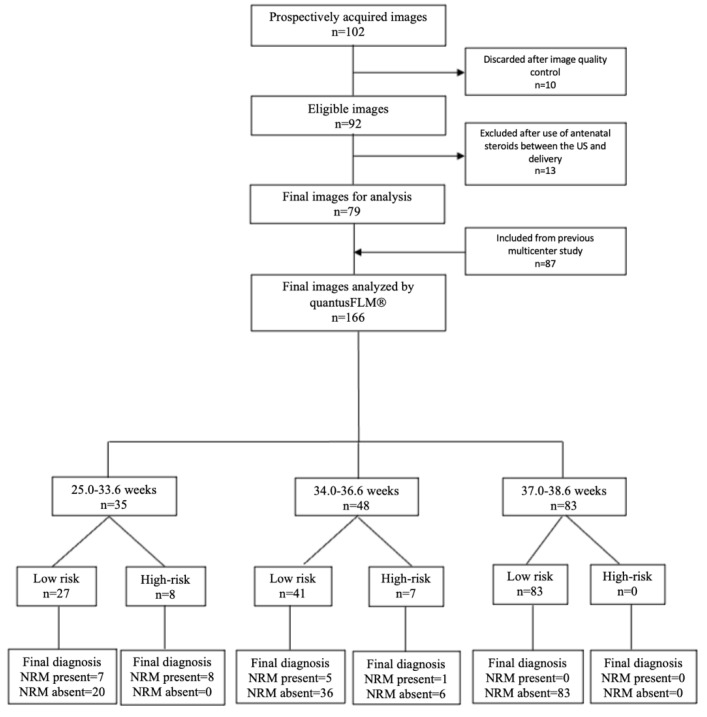
Flowchart of the eligible cases according to STARD guidelines.

**Figure 2 jcm-11-04895-f002:**
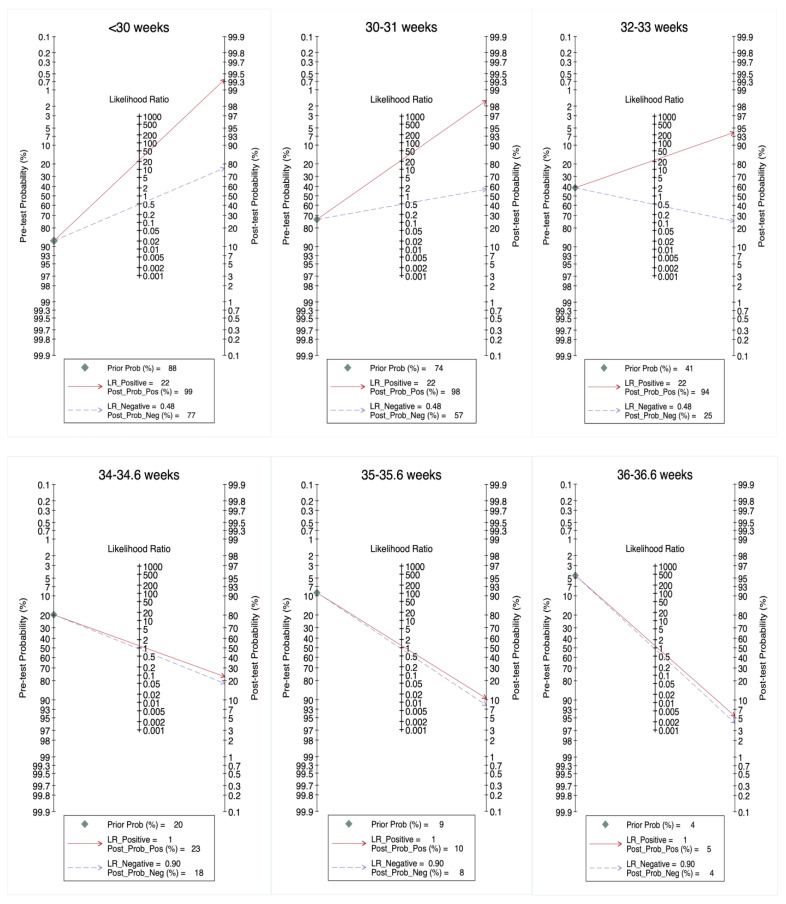
Fagan nomogram analysis to evaluate the clinical utility of the prediction of neonatal respiratory morbidity by quantusFLM^®^ in twin pregnancies. Pre-test probabilities of neonatal respiratory morbidity in twins were obtained from Clin Chim Acta 484 (2018) 293–297 for <30.0 up to 33.6 weeks [22] and from BMJ 2016;354: i4353 for 34.0 up to 36.6 weeks of gestation [3].

**Table 1 jcm-11-04895-t001:** Baseline characteristics of women included in the study.

		Gestational Age at US Scan, Weeks		
	Total (*n* = 105)	25.0–33.6 (*n* = 21)	34.0–36.6 (*n* = 32)	37.0–38.6 (*n* = 52)
Maternal age, mean (SD)	34.6 (4.3)	33.8 (4.8)	34.4 (3.6)	35.0 (4.4)
Nulliparity, *n* (%)	72 (68.6)	14 (66.7)	23 (71.9)	35 (67.3)
BMI, median (IQR)	22.7 (20.8–24.7)	22.4 (21.0–25.0)	22.7 (19.8–23.7)	23.0 (21.2–25.5)
Ethnicity, *n* (%)				
Caucasian, *n* (%)	87 (83.7)	14 (66.7)	30 (93.8)	43 (84.3)
Black	0 (0)	0 (0)	0 (0)	0 (0)
Asian	3 (2.9)	0 (0)	2 (3.1)	2 (3.9)
Hispanic	9 (8.7)	6 (28.6)	0 (0)	3 (5.9)
Other	5 (4.7)	1 (4.7)	1 (3.1)	3 (5.9)
Chorionicity, *n* (%)				
Dichorionic	61 (66.3)	8 (40.0)	17 (63.0)	36 (80.0)
Monochorionic	31 (33.7)	12 (60.0)	10 (37.0)	9 (20.0)
Antenatal steroids, *n* (%)	32 (31.4)	21 (100)	8 (25.0)	4 (7.7)
Relevant maternal–fetal conditions, *n* (%)				
IVF	38 (36.2)	5 (23.8)	13 (40.6)	20 (38.5)
Preeclampsia	15 (14.3)	5 (23.8)	9 (28.1)	1 (2.0)
IUGR ^a^	16 (15.2)	4 (19.1)	9 (28.1)	3 (5.8)
Diabetes	9 (8.6)	4 (19.1)	2 (6.3)	3 (5.8)
Preterm labor/PPROM	15 (14.3)	9 (42.9)	5 (15.6)	1 (2.0)

Data are presented as the mean (SD: standard deviation), number (%: percentage), or median (IQR: interquartile range) when appropriate. US: ultrasound; BMI: body mass index; IUGR: intrauterine growth restriction; IVF: in vitro fertilization; PPROM: preterm premature rupture of membranes. ^a^ IUGR in 16 mothers corresponds to 23/166 (13.9%) fetuses.

**Table 2 jcm-11-04895-t002:** Perinatal and neonatal outcomes.

		Gestational Age at US Scan, Weeks		
	Total (*n* = 166)	25.0–33.6 (*n* = 35)	34.0–33.6 (*n* = 48)	37.0–38.6 (*n* = 83)
GA at delivery, median (IQR)	36.8 (34.6–37.5)	33 (32.1–33.4)	36.3 (35.4–36.4)	37.5 (37.2–38.1)
US-to-delivery time days, mean (SD)	0.7 (0.8)	0.9 (0.7)	0.8 (0.8)	0.5 (0.8)
Mode of delivery, *n* (%)				
Spontaneous vaginal delivery	34 (20.5)	6 (17.1)	16 (33.3)	12 (14.5)
Operative vaginal delivery	9 (5.4)	0 (0)	5 (10.4)	4 (4.8)
Non-elective cesarean delivery	26 (15.7)	7 (20.0)	6 (12.5)	13 (15.7)
Elective cesarean delivery	97 (58.4)	22 (62.9)	21 (43.8)	54 (65.0)
Birthweight, mean (SD)	2313 (510)	1665 (333)	2199 (359)	2653 (319)
Female sex, *n* (%)	89 (53.9)	18 (51.4)	30 (62.5)	41 (50.0)
Apgar at 5 min < 7, *n* (%)	0 (0)	0 (0)	0 (0)	0 (0)
pH UA < 7.10, *n* (%)	3 (1.8)	0 (0)	3 (6.3)	0 (0)
Phototherapy, *n* (%)	26 (15.7)	14 (40)	10 (20.8)	2 (2.4)
NICU admission, *n* (%)	36 (21.7)	26 (74.3)	9 (18.8)	1 (1.2)
NICU length of stay, mean (SD)	15.3 (13.3)	17.9 (14.2)	8.8 (6.2)	1 (1.0)
Neonatal death, *n* (%)	0 (0)	0 (0)	0 (0)	0 (0)
Characteristics of the respiratory morbidity and support, *n* (%)				
Neonatal respiratory morbidity	21 (12.7)	15 (42.9)	6 (12.5)	0 (0)
Respiratory distress syndrome	8 (4.8)	8 (22.9)	0 (0)	0 (0)
Transient tachypnea	13 (7.8)	7 (20.0)	6 (12.5)	0 (0)
Respiratory support	21 (12.7)	16 (45.7)	5 (10.4)	0 (0)
Oxygen therapy ≥ 40%	9 (5.4)	7 (20.0)	2 (4.2)	0 (0)
CPAP	20 (12.1)	15 (42.9)	5 (10.4)	0 (0)
NIV/BiPAP	6 (3.6)	4 (11.4)	2 (4.2)	0 (0)
Invasive ventilation	5 (3.0)	5 (14.3)	0 (0)	0 (0)
High-frequency ventilation	0 (0)	0 (0)	0 (0)	0 (0)
Surfactant use	7 (4.2)	7 (20.0)	0 (0)	0 (0)

Data are presented as the mean (SD: standard deviation) or number *n* (%: percentage) when appropriate. GA: gestational age; US: ultrasound; pH UA: pH umbilical artery; NICU: neonatal intensive care unit; CPAP: continuous positive airway pressure; NIV/BiPAP: non-invasive/bilevel positive airway pressure.

**Table 3 jcm-11-04895-t003:** Performance of quantusFLM in predicting neonatal respiratory morbidity in twin pregnancies.

		Gestational Age at US Scan, Weeks		
	Total (*n* = 166)	25.0–33.6 (*n* = 35)	34.0–36.6 (*n* = 48)	37.0–38.6 (*n* = 83)
Neonatal respiratory morbidity, *n* (%)	21 (12.7)	15 (42.9)	6 (12.5)	0 (0)
Sensitivity, % (95% CI)	42.9 (24.5–63.5)	53.1 (30.1–75.2)	16.7 (3.0–56.4)	n/a
Specificity, % (95% CI)	95.9 (91.3–98.1)	97.6 (83.9–100)	85.7 (72.2–93.3)	99.9 (95.6–100)
True positives (*n*)	9	8	1	0
True negatives (*n*)	139	20	36	83
False positives (*n*)	6	0	6	0
False negatives (*n*)	12	7	5	0
Positive predictive value, % (95% CI)	60.0 (35.7–80.2)	94.4 (67.6–100)	14.3 (2.6–51.3)	n/a
Negative predictive value, % (95% CI)	92.1 (86.6–95.4)	73.4 (55.3–86.8)	87.8 (75.0–97.8)	99.9 (95.6–100)
Accuracy, % (95% CI)	89.2 (83.5–93.0)	78.4 (64.1–90.0)	77.1 (63.5–86.7)	n/a
F1-Score (%)	50.0	68.0	15.4	n/a
Positive likelihood ratio (95% CI)	10.4 (4.1–26.1)	22.3 (19.0–26.2)	1.2 (0.2–8.1)	n/a
Negative likelihood ratio (95% CI)	42.9 (24.5–63.5)	53.1 (30.1–75.2)	16.7 (3.0–56.4)	n/a

Data are presented as the percentage (%) when appropriate. CI: confidence interval. Continuity correction factor of 0.5 for cells with cero values. n/a: not applicable. US: ultrasound.

## Data Availability

The data presented in this study are available on request from the corresponding author. The data are not publicly available due to restrictions according to patient privacy regulations.

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
