# Peer review of "Prediction of Neonatal Respiratory Morbidity Assessed by Quantitative Ultrasound Lung Texture Analysis in Twin Pregnancies"

_jcm, 2022, doi:10.3390/jcm11164895_

Round 1

Reviewer 1 Report

Thank you for the opportunity to review so interesting, well designed and clinically useful manuscript.

A few suggestions. 

1. Flowchart. I suggest to describe 83 cases above 37 weeks in the same manner as other cases.

2. It would be useful to add the information on prevalence of NRM in 25-36,6 weeks (83 cases) of gestation in the text in the part Results or Discussion.

3. I calculated that 53 women at 25-36,6 weeks of gestation were included in the study for the final analysis. 53 women - 106 imagines of the every fetus. The authors presented 83 neonatal case in tis group of gestation. It would be very useful, especially for clinicians, to explain what were the main problems that all pairs  of twins are not included - image quality problems, it were usually with the Fetus A, or Fetus B; it depends on fetal lie or position or other.

Reviewer 2 Report

In this study by Moreno-Espinosa et al (Manuscript ID: jcm-1836095) entitled “Prediction of neonatal respiratory morbidity assessed by quantitative ultrasound lung texture analysis in twin pregnancies”, the authors evaluated the performance of quantitative ultrasound of the fetal lung texture analysis as predictor of neonatal respiratory morbidity in twin pregnancies. The results from this study showed an overall good performance of this technique with high specificity, negative predicted value, and accuracy.

This is a well-written study and of significant clinical interest. However, I have some comments:

Material and methods

·      Paragraph 2.2. Images and more specifically Regions of Interest were taken from both fetuses or only from one of them? This should be clarified, explaining at the same time why the numbers of images and mothers are equal, as shown in Table 1.

·      Paragraph 2.3, L 115-121. The diagnosis of RDS and TTN is typically based on the chest X-ray finding and are not related to the clinical signs and symptoms. Neonates with TTN may also have grunting, retractions, etc. Therefore, this paragraph describing RDS/TPN definitions should be corrected.

Discussion

·      The paragraph on the prenatal steroid administration (L 254-277) should be shortened. This is not the point of the study.

Table 2.

·      Please write “Respiratory distress syndrome” instead of “Respiratory distress”.

Table 3.

·      In my opinion, this table is kind of confusing because, the “34.0-36.6 group” (n=48) is not shown. Of course, there is the “a” sign explaining that “a83 cases of the sample correspond to the group > 37.0 weeks” but still this table seems incomplete to me. Non applicable could clarify the lack of cases with RDS/TTN precluded the calculation of various performance indicators.

Figure 1.

·      The box with the 83 cases of term neonates should be at the same line as the other two groups, in order for the reader to get a more clear image of the cases included in the study.

Figure 2.

·      References shown in the figure such as Clin Chim Acta 484 (2018) 293-297, and BMJ 2016;354: i4353 219 should be included in the references list and noted accordingly, e.g. [X].
